# A standardized method for plasma extracellular vesicle isolation and size distribution analysis

J. Nathaniel Diehl[ORCID][1]*, Amelia Ray[ORCID][1], Lauren B. Collins[1], Andrew Peterson[2,3], Kyle C. Alexander[2,3], Jacob G. Boutros[4], John S. Ikonomidis[2,3], Adam W. Akerman[2,3]

1 University of North Carolina School of Medicine, Chapel Hill, North Carolina, United States of America, 2 Department of Surgery, University of North Carolina–Chapel Hill, Chapel Hill, North Carolina, United States of America, 3 Division of Cardiothoracic Surgery, University of North Carolina–Chapel Hill, Chapel Hill, North Carolina, United States of America, 4 Campbell University School of Osteopathic Medicine, Lillington, North Carolina, United States of America

* natediehl@gmail.com

**Data Availability Statement:** All relevant data are within the manuscript and its Supporting Information files.

## Abstract

The following protocol describes our workflow for isolation and quantification of plasma extracellular vesicles (EVs). It requires limited sample volume so that the scientific value of specimens is maximized. These steps include isolation of vesicles by automated size exclusion chromatography and quantification by tunable resistive pulse sensing. This workflow optimizes reproducibility by minimizing variations in processing, handling, and storage of EVs. EVs have significant diagnostic and therapeutic potential, but clinical application is limited by disparate methods of data collection. This standardized protocol is scalable and ensures efficient recovery of physiologically intact EVs that may be used in a variety of downstream biochemical and functional analyses. Simultaneous measurement quantifies EV concentration and size distribution absolutely. Absolute quantification corrects for variations in EV number and size, offering a novel method of standardization in downstream applications.

## Introduction

Extracellular vesicles (EVs) are a heterogeneous class of lipid bound particles released by most cell types. EVs are involved with a wide range of biological processes, including cell-to-cell communication, immune system modulation, and extracellular matrix remodeling [1]. EVs carry a variety of cargo and alterations in their number and composition likely indicate pathology [2–6]. Because of these clear roles in homeostasis and pathology, EVs have recently been evaluated for their diagnostic potential.

While EVs alone may possess significant diagnostic potential, current research methodologies face various technical challenges, these include: standardization of isolation methods; optimization of EV cargo analysis; accurate size-based quantification [7–9]. Although EVs can be isolated from all bodily fluids and from tissues, blood provides the richest source [10]. Separation of EV-containing plasma from whole blood allows for effective long-term storage. Plasma

**Funding:** Research reported in this publication was supported by the University of North Carolina at Chapel Hill and the National Heart, Lung, and Blood Institute (NHLBI) of the National Institutes of Health: R01HL102121, R21HL148363, R56HL161454 (J.S. Ikonomidis); 2021 UNC Idea Grant: Development of a Liquid Biopsy Based Diagnostic Test for Assessing Thoracic Aortic Aneurysm Disease (A.W. Akerman). The content is solely the responsibility of the authors and does not necessarily represent the official views of UNC or the National Institutes of Health. The funders had and will not have a role in study design, data collection and analysis, decision to publish, or preparation of the manuscript.

**Competing interests:** The authors have declared that no competing interests exist.

samples offer, in the form of liquid biopsy, a relatively non-invasive means to detect disease-related biomarkers in EVs, making them an optimal target for investigations using large study cohorts and biobanks [11]. To avoid confounders resulting from blood collection, processing, and storage, the International Society of Extracellular Vesicles (ISEV) taskforce has taken steps toward standardization [12]. Thus, this protocol has selected to use blood plasma collected in the presence of ethylenediaminetetraacetic acid (EDTA) and stored at -80˚C. EDTA has less effects on downstream biochemical applications in comparison to other anti-coagulants, such as heparin or sodium citrate [13–17].

Unlike other biofluids, plasma EV yield (concentration and cargo) is not adversely affected by freeze-thaw and long-term storage of plasma at -80˚C [18–20]; however, long term storage leads to decreases in EV-associated acetylcholine-esterase (AChE) activity [21]. AChE is a plasma membrane protein incorporated into EVs during biogenesis [22]. In the past, AChE enzymatic activity was used as a surrogate measure of EV concentration [23]. However, recently AChE activity was demonstrated to not be a universal marker of EVs and cannot be used to quantitate EVs reliably, except in specific cell systems [24].

Quantification of EVs from plasma is affected by the presence of similarly-sized contaminants [25]. Differential centrifugation protocols have been used many times, but this practice is not viable for several reasons having to do with the stress applied to vesicles by centrifugal forces [26]. These problems extend to size determination and characterization as well. Accurate determination of size distribution of EVs is paramount when choosing the best method of precise measurement. Size characterization is fraught with challenges because EVs are very small and heterogenous. A "Traceable size determination" is necessary, wherein the requisite measurement is based off the SI unit "metre" [27]. Once established, EVs usefulness can be fully realized: development of reference materials, calibration of other detection methods, and the facilitation of true standardization.

Varying nomenclature signifying many different EV sub-populations can be found in the literature [25]. These can be attributed to cellular origin, biogenesis, size, function, cargo, or membrane markers. EVs are divided into two groups: exosomes and microvesicles. Exosomes are commonly reported to be 30–120 nm in size and are released from multivesicular endosomes following fusion with the plasma membrane [28]. In both function and size, microvesicles are extremely heterogeneous: their size ranges between 50 nm and 1 μm, and they shed into the extracellular milieu by the budding of the cell membrane [28]. Of diverse density and morphology, with new subpopulations (*e.g.* exomeres, <35 nm) still being (and to be) discovered, EVs are more complex than we know today and further investigations are sure to find yet more characteristics and subgroups [29].

This protocol focuses on isolation and quantification of plasma EVs 30–300 nm in diameter. This size range is the most reported in the literature for the exosome/microvesicle population and can be isolated in a single preparation; it also excludes larger apoptotic bodies, oncosomes, and smaller exomeres [30, 31]. While it may seem problematic at first glance that both exosomes and microvesicles populate the specified size range, that fact does not impact the purpose of this protocol, which is to assert a method of standardized, reproducible measurement of particles between 30–300 nm. This is necessary for reproducible measurements and downstream applications.

When size exclusion chromatography (SEC) is performed with automated fraction collectors, it not only isolates vesicles with high levels of purity and recovery efficiency, it also scales with each additional collector [32]. Other available methods, such as asymmetric flow field-flow fractionation (AF4), tangential flow fractionation (TFF), differential centrifugation, and precipitation are not scalable in this fashion. These methods are not especially precise or produce a low yield making downstream biochemical analyses problematic [25]. Polyethylene

glycol (PEG) precipitation results in residue on EV surface which artificially increases particle size and may interfere with the binding of ligands to EV surface receptors [31, 33]. Moreover, the extraneous forces applied during centrifugation often damage EVs such that they are no longer useful in this application [26]. Additionally, these methods are particularly labor intensive and have been demonstrated to isolate a homogenous mixture of other vesicle types (*e.g.* exomeres and apoptotic bodies), resulting in variability [34].

By contrast automated SEC isolates a specific size range of vesicles and is not dependent on density [35]. In this protocol, molecules (<35 nm) are slowed because they enter the pores of the stationary phase. Larger particles that cannot enter the pores flow around the resin and are eluted from the column. Molecules and small particles that enter the pores have longer retention times and elute later. SEC and the various ways we plan to leverage this technology address the limitations of EV isolation by microfluidics, precipitation, and centrifugation.

Several methods of EV quantification exist. These include resistive pulse sensing, microscopy (scanning/transmission electron microscopy, and atomic force microscopy), dynamic light scattering (DLS), nanoparticle tracking analysis (NTA), flow cytometry, and small-angle X-ray scattering (SAXS) [27]. Microscopy, flow cytometry, NTA and DLS, are constrained in absolute quantitative abilities [36]. While resistive pulse sensing and SAXS quantify absolutely, SAXS requires a specific infrastructure only available at limited synchrotron radiation laboratories worldwide [27].

In this protocol, quantification of EVs is performed by tunable resistive pulse sensing (TRPS). Unlike other methods, TRPS is not dependent on the selection of measurement parameters, i.e. camera settings and detection threshold. TRPS provides highly precise and accurate measurements with a much higher level of resolution than NTA or dynamic light scattering [37]. This technology, which is based on the Coulter principle, is recognized as the most reliable and accurate method with sub nm precision [38, 39]. TRPS quantifies absolutely, whereas other commonly used light scattering techniques provide bulk estimates. TRPS measures EVs suspended in electrolyte on a particle-by-particle basis. TRPS simultaneously measures EV size and concentration. As particles individually pass through a nanopore it creates a blockade, the magnitude of the blockade is directly proportional to particle size while the frequency of the blockades determines concentration [39]. These components are all necessary for therapeutic and diagnostic development because quantification and dosage determinations are based on standardized inputs. Importantly, TRPS quantifies EVs rather than providing an estimate of concentration.

Myriad methods can analyze EVs and each may be appropriate for specialized applications [40–42]. For our purposes, however, a combination of automated SEC and TRPS works best. Because our protocol provides simultaneous measurements of absolute quantification and size distribution of EVs 30–300 nm, applications like flow cytometry, NTA, and DLS, are not appropriate as detection is often limited to particles greater than 100 nm (or 60 nm for NTA/ DLS) in diameter or do not provide absolute quantification [43, 44]. The uniqueness of this protocol lies in the way we leverage this combination of SEC and TRPS. Specifically, this protocol provides a method for reproducible quantification and is performed in a way that allows vesicles to remain intact throughout the measurement process, thus providing novel methods for normalization in any downstream application.

Limitations for the use of this protocol include access to equipment and supply. It also requires a properly trained technician. While specialized training is required, this method has proven to be consistent in data collection between individuals with different levels of laboratory experience. Nevertheless, this protocol allows for reproducible isolation and quantification of plasma derived EVs.

## Materials and methods

The protocols described in this peer-reviewed article are published on protocols.io (dx.doi.org/ 10.17504/protocols.io.3byl4jeb2lo5/v1, dx.doi.org/10.17504/protocols.io.ewov1ojnolr2/v1) and are included as S1 and S2 Files with this article.

### Immunoblotting

Following SEC isolation as described in the accompanying protocol (DOI dx.doi.org/10. 17504/protocols.io.3byl4jeb2lo5/v1), the first three EV fractions were combined (800 μl) and concentrated to 100 μl using a centrifugal filter with a 100 kDa molecular weight cut-off (UFC810024, Amicon). While we recognize some EVs may be lost during centrifugal concentration, this step allows 10 μg of protein to be loaded to improve resolution in western blotting. For whole cell lysate, 1 million HT1080 cells were lifted with TrypLE (12604021; Thermo Fisher Scientific), and pelleted. Halt Protease Inhibitor Cocktail (87785; ThermoFisher Scientific) was added to the pelleted HT1080 cells. As previously described by our group [45], a portion of each sample was taken for protein quantification with Pierce bicinchoninic acid protein assay kit (23225; Thermo Fisher Scientific). HT1080 cell lysate samples were diluted 1:1 in 2x Laemmli buffer, heated at 90°C for 10 minutes, and stored at -80°C. Concentrated EV fractions were stored at -80°C, and a portion of sample was diluted 1:1 in 2x Laemmli buffer and heated at 90°C for 10 minutes prior to loading equal amounts of total protein (10 μg) onto 4% to 15% Mini-PROTEAN TGX Stain-Free Gels (4568084; Bio-Rad Laboratories) and transferred onto nitrocellulose membranes (10600007, Cytiva). After blocking overnight in 5% BSA (97061–422; VWR) at 4°C, nitrocellulose membranes were incubated for 1 hour at room temperature in primary antisera for CD9 (ab92726, Abcam; 1:1000), CD68 (ab125212, Abcam; 0.5 μg/mL), CD81 (ab109201, Abcam; 1:1000), CD63 (ab134045, Abcam; 1:1000), GM130 (ab187514, Abcam; 1:1000), GAPDH (VPA00187, Bio-Rad Laboratories; 1:5000), MMP-14 (ab38971, Abcam; 1:2000), MMP-2 (ab92536, Abcam; 1:1000), TGFβ-1 (ab215715, Abcam; 1:1000), and TIMP-2 (ab180630, Abcam; 1:500). Membranes were washed in TBST, incubated for 1 hour at room temperature in horseradish peroxidase labeled Goat-Anti-Rabbit secondary antibody (GtxRb-003-EHRPC; Immunoreagent Inc, Raleigh, NC; EVs and cells: 1:5000), and washed in TBST. Chemiluminescent substrate activation was performed with Super Signal West Pico PLUS following manufacturer's protocols (34580; Thermo Fisher Scientific). Full blot/gel images are available in S1 Fig.

## Expected results

Here we detail a multipart protocol for the isolation and measurement of extracellular vesicles from plasma. Our protocol describes reproducible methods for isolation and storage, extracellular vesicle purification with SEC, and simultaneous quantification of EV concentration and size distribution with TRPS. Our aim is to delineate the most effective and efficient ways to apply SEC and TRPS techniques to EV samples i.e., what parameters to prioritize and any potential pitfalls to be avoided. We offer detailed troubleshooting steps derived from our experiences, and we show that our results are highly reproducible both temporally and between instrument operators. While our purpose is not to demonstrate the biological significance of our findings, we do show that our protocol has the potential to identify biomarker features in human plasma.

This protocol was optimized and refined through learned experience during the early stages of our work. Importantly, our protocol offers specific guidance on the peripheral blood draw component of human plasma isolation. This step can vary widely dependent upon the

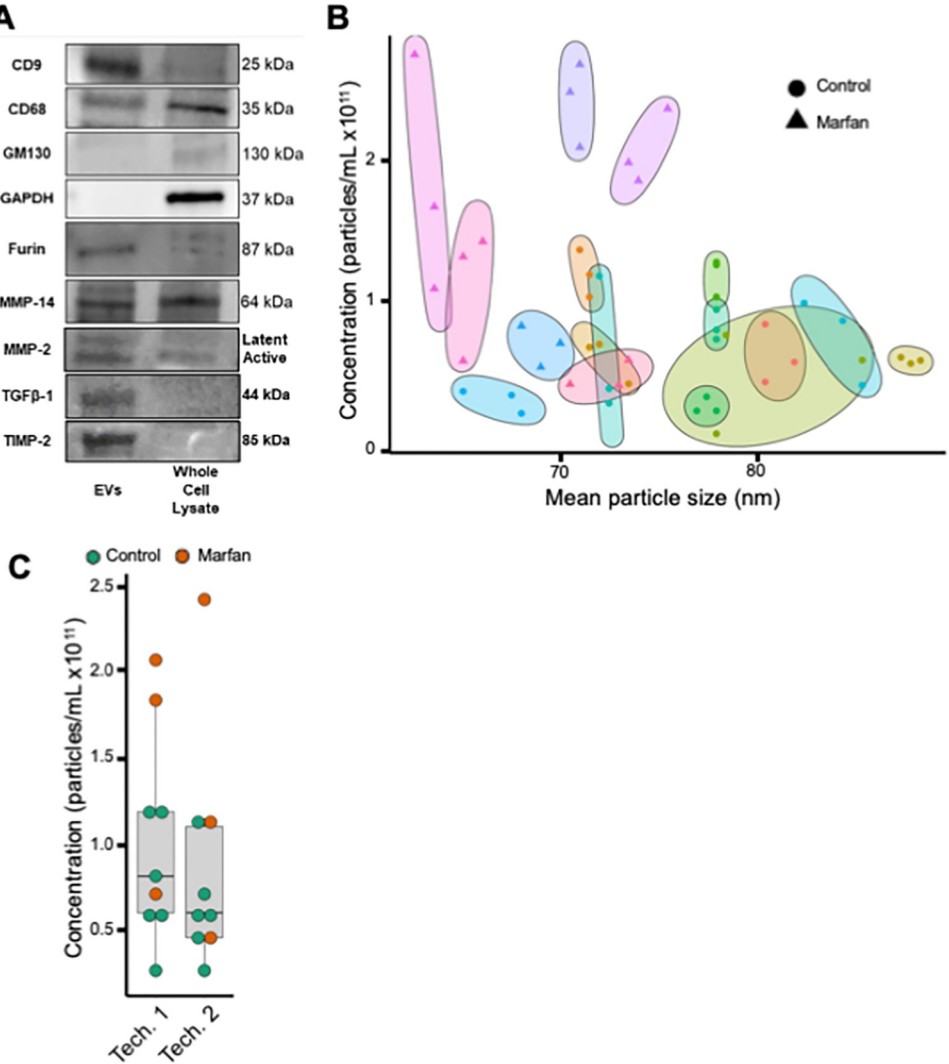

**Fig 1. EV isolation and technical reproducibility following SEC isolation and TRPS measurement.** (A) Western blot comparing prototypical markers and investigational targets in isolated EVs and whole cell lysate from HT1080 cells. (B) Dot plot comparing mean particle size (x-axis) and EV concentration (y-axis) among technical replicates of human plasma. Technical replicates are grouped by color and a correspondingly shaded ellipse. (C) Comparison of sample concentration (y-axis) by technician operating the TRPS instrument. EV concentration detected by technician 1 (median 0.81 x $10^{11}$ particles/mL, interquartile range 0.59–1.19 x $10^{11}$ particles/mL) and technician 2 (median 0.60 x $10^{11}$ particles/mL, interquartile range 0.45–1.10 x $10^{11}$ particles/mL, p = 0.606) are displayed separately. A Student's t-test was used to test for significance.

institutional and patient context. Our specific recommendations adhere to the guidelines established by the International Society of Extracellular Vesicles (ISEV) taskforce [12].

Before EVs can be quantified, we first determined whether the SEC method was able to robustly isolate EVs. Following isolation of EVs from plasma, markers for EV subtypes can be confirmed with various protein analyses, such as western blotting (Fig 1A, S1 Fig). For exosomes and microvesicles, markers clusters of differentiation 9 (CD9), 63 (CD63), 68 (CD68), and 81 (CD81) are recommended [34]. Moreover, EVs must be negative for the endosomal marker Golgin subfamily A member 2 (GM130) and glyceraldehyde-3-phosphate dehydrogenase (GAPDH) [46]. In adherence with the MISEV2018 guidelines, this must be performed

prior to use of EVs in all experimentation [34]. Furthermore, EVs remain positive for extracellular matrix remodeling proteins such as, matrix metalloproteinase 14 (MMP-14), matrix metalloproteinase 2 (MMP-2), transforming growth factor beta-1 (TGFβ-1), and tissue inhibitor of metalloproteinase 2 (TIMP-2). SEC allows for isolation of EVs that can be used in a variety of different downstream applications. These applications range from biochemical to functional analyses, such as matrix degradation assays or gene transfer in vitro or in vivo. Accordingly, the importance of accurate quantification is paramount for normalization of the above downstream applications and diagnostic development.

During protocol development, we also addressed the question of whether this protocol can reliably and reproducibly measure EV concentration and size distribution. We measured three technical replicates of the same patient plasma and found that EV features were highly consistent across replicates (Fig 1B, S1 Table). Some variability existed between measurements of the same sample. Most technical replicates, however, were more similar to one another than to

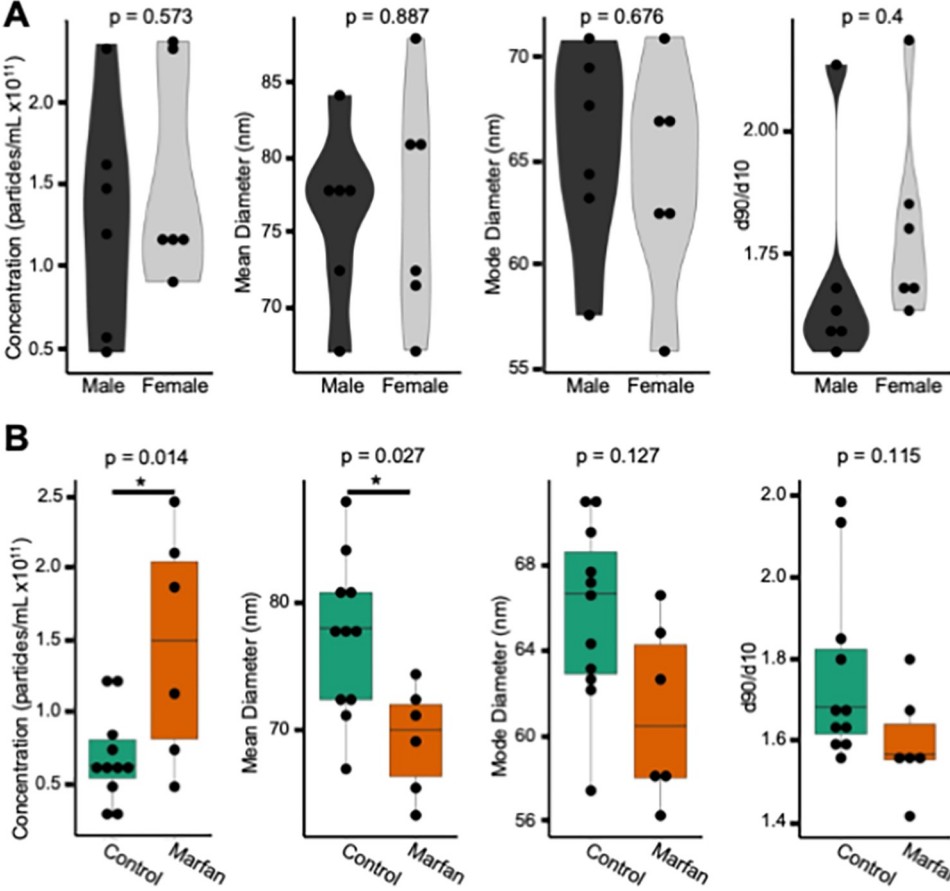

**Fig 2. EV concentration and size isolated from human plasma are dependent on disease state.** (A) EVs isolated from control patient plasma (n = 12) were measured for concentration (male 0.77 x 10¹¹ particles/mL, female 0.63 x 10¹¹ particles/mL), mean diameter of EV particle (male 73.33 nm, female 71.83), mode diameter of EV particles (male 64.58 nm, female 62.67 nm), and d90/d10 ratio (male 1.59, female 1.74). Differences between males (n = 6) and females (n = 6) were assessed with a Student's t-test. Corresponding p-values are displayed, and median sample values are detailed here. (B) EVs isolated from control plasma (n = 12) and Marfan patient plasma (n = 6) were measured for concentration (control 0.60 x 10¹¹ particles/mL, Marfan 1.47 x 10¹¹ particles/mL), mean diameter of EV particle (control 77.75 nm, Marfan 69.92 nm), mode diameter of EV particles (control 65.50 nm, Marfan 60.50 nm), and d90/d10 ratio (control 1.68, Marfan 1.57). Differences between control and Marfan patients were assessed with a Student's t-test. Corresponding p-values are displayed, and median sample values are detailed here.

other plasma samples. We recommend using at least two technical replicates during measurements of EVs, but in most cases one technical replicate will suffice. We also tested whether the instrument operator would significantly influence the sample measurements. Each operator measured six control samples and three pathological specimens (patients with Marfan syndrome-related thoracic aortic aneurysms); we found no significant difference in EV concentrations (Fig 1C). Both operators followed the protocols described herein, which we speculate contributed to the remarkable consistency in the sample measurements. Additionally, our protocol details a hierarchical approach to troubleshooting the TRPS instrument along with practical advice regarding time management and experimental setup. We also take users through the data analysis and quantification process in a step-by-step manner. While our methods are robust for the instrument set up that we describe, it would be interesting to see if these methods contribute to similar reproducibility for results collected on other TRPS instruments.

We utilized the techniques described here to address a clinically meaningful question. In our system, we measure EV concentration and size in the plasma of healthy control patients and in Marfan patients who possessed a known thoracic aortic aneurysm. Previously, we have demonstrated that increased aortic wall tension, as seen in aneurysm disease, leads to elevated aortic fibroblast secretion of EVs [47]. The primary cardiovascular complication associated with Marfan Syndrome is thoracic aortic aneurysm. Therefore, we are interested in determining whether EV concentration and size distribution can be leveraged as a diagnostic marker for aneurysm disease. We show that EV concentration does not vary significantly between sexes (Fig 2A). Next, we found that Marfan patients have higher concentrations of EVs and smaller mean diameters of EVs than the control patients (Fig 2B). The use of SEC and TRPS for the measurement of EVs in plasma yielded significant differences between healthy and disease states. More work is needed to understand the significance of this finding in the context of the disease pathophysiology. Additionally, development of methods for quantification and characterization of nucleic acid, fatty acid, and protein content of EVs remains underway.

## Supporting information

**S1 File. Human sample processing and isolation of extracellular vesicles with size exclusion chromatography.** Step-by-step protocol, also available on protocols.io.
(PDF)

**S2 File. Measurement of extracellular vesicles with tunable resistive pulse sensing.** Step-by-step protocol, also available on protocols.io.
(PDF)

**S1 Fig. Full gel/blot images accompanying Fig 1A.** Each column represents visualization of total protein loaded onto the gel (Activation), transferred onto the nitrocellulose (Transfer), or specific target detection (Immunodetection). Bio Rad TGX gels were used for protein separation by molecular weight. These gels contain a trihalo compound which modifies tryptophan residues in protein samples by a covalent modification. When exposed to ultraviolet (UV) excitation, a fluorescence signal is visualized representing total protein in both the gel (Activation) and on the nitrocellulose (Transfer). Chemiluminescent immunodetection is recorded by ChemiDoc Imaging System for each antibody exposure A) CD9 B) CD68 C) CD81 D) CD63 E) GM130 F) GAPDH G) MMP-14 H) MMP-2 I) TGFβ J) TIMP-2. Gel order: Ladder, EVs, whole cell lysate. The yellow box indicates the cropped region included in Fig 1A.
(PDF)

**S1 Table. Formatted TRPS data output.** Data table containing TRPS data with technical replicates.
(CSV)

## Acknowledgments

The authors thank Izon Science Ltd. for their help with initial instrument set up and protocol optimization.

## Author Contributions

**Conceptualization:** J. Nathaniel Diehl, Adam W. Akerman.

**Data curation:** J. Nathaniel Diehl, Amelia Ray, Jacob G. Boutros.

**Formal analysis:** J. Nathaniel Diehl, Amelia Ray.

**Funding acquisition:** J. Nathaniel Diehl, John S. Ikonomidis, Adam W. Akerman.

**Investigation:** J. Nathaniel Diehl, Amelia Ray, Lauren B. Collins, Jacob G. Boutros, Adam W. Akerman.

**Methodology:** J. Nathaniel Diehl.

**Project administration:** Andrew Peterson, John S. Ikonomidis, Adam W. Akerman.

**Resources:** John S. Ikonomidis, Adam W. Akerman.

**Software:** J. Nathaniel Diehl.

**Supervision:** John S. Ikonomidis, Adam W. Akerman.

**Validation:** J. Nathaniel Diehl, Andrew Peterson.

**Visualization:** J. Nathaniel Diehl, Adam W. Akerman.

**Writing – original draft:** J. Nathaniel Diehl, Andrew Peterson, Kyle C. Alexander, Adam W. Akerman.

**Writing – review & editing:** J. Nathaniel Diehl, Andrew Peterson, Kyle C. Alexander, Jacob G. Boutros, Adam W. Akerman.

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
