## [Decision Letter · Decision Letter 0]

19 Oct 2022

PONE-D-22-26362A standardized method for plasma extracellular vesicle isolation and size distribution analysisPLOS ONE

Dear Dr. Diehl,

Thank you for submitting your manuscript to PLOS ONE. After careful consideration, we feel that it has merit but does not fully meet PLOS ONE’s publication criteria as it currently stands. Therefore, we invite you to submit a revised version of the manuscript that addresses the points raised during the review process.

We look forward to receiving your revised manuscript.

Kind regards,

Vibhuti Agrahari

Academic Editor

PLOS ONE

Reviewers' comments:

Reviewer's Responses to Questions

**Comments to the Author**

1. Does the manuscript report a protocol which is of utility to the research community and adds value to the published literature?

Reviewer #1: Yes

Reviewer #2: Yes

2. Has the protocol been described in sufficient detail?

To answer this question, please click the link to protocols.io in the Materials and Methods section of the manuscript (if a link has been provided) or consult the step-by-step protocol in the Supporting Information files.

The step-by-step protocol should contain sufficient detail for another researcher to be able to reproduce all experiments and analyses.

Reviewer #1: Yes

Reviewer #2: Yes

3. Does the protocol describe a validated method?

Reviewer #1: No

Reviewer #2: Yes

4. If the manuscript contains new data, have the authors made this data fully available?

Reviewer #1: N/A

Reviewer #2: Yes

**5. Is the article presented in an intelligible fashion and written in standard English?**

Reviewer #1: Yes

Reviewer #2: Yes

6. Review Comments to the Author

Reviewer #1: In this manuscript, the authors describe protocols for preparing plasma samples to be subsequently processed by automated size exclusion chromatography and tunable resistive pulse sensing to determine the concentration and size distribution of EVs in plasma.

Protocols of this kind can be helpful in particular for applications of EVs as biomarkers of diseases using small blood plasma volume or for analytic purposes to characterize further and study the functional activities of the isolated EV e.g., in cellular models.

The authors provide substantial evidence supporting the reliability of the methods they describe. The possibility to scale up the procedures is more arguable, as other types of fractionation methods using other commercial size exclusion resins and hardware would instead be used.

One primary concern I have with the manuscript is its questionable novelty as compared to other (non-cited) publications (even though I admit they were not all written as protocols) describing the way how to process blood samples and isolate (including by q-EVs) and analyze (including by TRPS) the EVs, e.g., https://pubmed.ncbi.nlm.nih.gov/20831623/;
https://www.ahajournals.org/doi/full/10.1161/CIRCRESAHA.117.309417;
https://www.cell.com/cell-reports-methods/pdfExtended/S2667-2375(21)00103-X). For instance, some of these papers do also provide "specific guidance on the peripheral blood draw component of human plasma isolation" and use q-EV SEC and TRPS. What is unique in the current manuscript that has not been highlighted in previous publications?

Regarding the validation of the procedures, I would also be interested to know why other methods like NTA and DLS have not been used to validate the isolation and characterization methods obtained by TRPS.

Other comments:

Line 72: isn't ---- > is not

Line 86: is it 10um that is meant? Or rather 1um?

Lines 117-118: NTA is listed twice

Lines 125 and below: It is debatable to claim that TRPS is the most reliable and accurate method with sub-nm precision, and is superior to the more commonly used light scattering techniques. TRPS has, like other methods, limitations for heterogeneous EV populations, and the data obtained are directly impacted by the type of membrane selected to avoid clogging. Most people would think combining techniques is suitable for characterizing heterogeneous EV populations. Therefore I do not see the need to promote TRPS (or other Izon products) in this manuscript that describes protocols.

Lines 149-150: is there any risk of losing EV on the membrane of the centrifugal filter?

Reviewer #2: Diehl and colleagues submitted a manuscript of protocol to explain the standardized method for the characterization of EVs in plasma.

This manuscript describes detailed procedures of plasma collection, EVs isolation, and sample procedures. In this paper, the explanation of the rationale for the selection of methods (SEC and TRPS). This paper is well-organized and has a logical basis.

If some requirements in the following comments are modified, this paper is going to be able to clear information for readers.

Minor comments.

1. Please add the information of reference at (DOI xx) line 148.

2. Spacing between number and unit (lines 84 and 86).

3. Add unit at diameter (line 269 (Marfan 69.92))

4. In terms of utilization of this technique, Marfan syndrome was selected to show the difference in the number and size distribution of EVs in plasma. Please add the physiological background information to explain why this syndrome could be a model as a diagnostic target.

7. PLOS authors have the option to publish the peer review history of their article (what does this mean?). If published, this will include your full peer review and any attached files.

Reviewer #1: No

Reviewer #2: No

---

## [Author Response · Author response to Decision Letter 0]

13 Dec 2022

We thank the reviewers for their thoughtful comments. We had addressed each one directly and have provided both a response and the amended manuscript text. This process improved the overall quality of our submission immensely. 

 We formatted the font size, line spacing, headings/subheadings, and placement of the figure legends to adhere to the specifications provided in the above links.

2. PLOS ONE now requires that authors provide the original uncropped and unadjusted images underlying all blot or gel results reported in a submission’s figures or Supporting Information files. 

 We have included all full blot images as specified by the editors. This has been noted in the cover letter. 

3. In your Data Availability statement, you have not specified where the minimal data set underlying the results described in your manuscript can be found. 

 We have included the full data sets in the supplement as specified by the editors. This has been noted in the cover letter.

Comments to the Author

1. Does the manuscript report a protocol which is of utility to the research community and adds value to the published literature?

Reviewer #1: Yes

Reviewer #2: Yes

2. Has the protocol been described in sufficient detail?

To answer this question, please click the link to protocols.io in the Materials and Methods section of the manuscript (if a link has been provided) or consult the step-by-step protocol in the Supporting Information files.

The step-by-step protocol should contain sufficient detail for another researcher to be able to reproduce all experiments and analyses.

Reviewer #1: Yes

Reviewer #2: Yes

3. Does the protocol describe a validated method?

Reviewer #1: No

Reviewer #2: Yes

 The protocol describes a validated method. Validation data is located in figure 2.

4. If the manuscript contains new data, have the authors made this data fully available?

Reviewer #1: N/A

Reviewer #2: Yes

 Validation data is in figure 2. Additionally, all data is now located in the supplement.

5. Is the article presented in an intelligible fashion and written in standard English?

Reviewer #1: Yes

Reviewer #2: Yes

6. Review Comments to the Author

Reviewer #1: In this manuscript, the authors describe protocols for preparing plasma samples to be subsequently processed by automated size exclusion chromatography and tunable resistive pulse sensing to determine the concentration and size distribution of EVs in plasma.

Protocols of this kind can be helpful in particular for applications of EVs as biomarkers of diseases using small blood plasma volume or for analytic purposes to characterize further and study the functional activities of the isolated EV e.g., in cellular models.

The authors provide substantial evidence supporting the reliability of the methods they describe. The possibility to scale up the procedures is more arguable, as other types of fractionation methods using other commercial size exclusion resins and hardware would instead be used.

 Thank you for recognizing the importance of these protocols. We have clarified how we plan to leverage automation to address concerns of scalability. The following statement has been added to the introduction.

“When size exclusion chromatography (SEC) is performed with automated fraction collectors, it not only isolates vesicles with high levels of purity and recovery efficiency, it also scales with each additional collector [32]. Other available methods, such as asymmetric flow field-flow fractionation (AF4), tangential flow fractionation (TFF), differential centrifugation, and precipitation are not scalable in this fashion.”

One primary concern I have with the manuscript is its questionable novelty as compared to other (non-cited) publications (even though I admit they were not all written as protocols) describing the way how to process blood samples and isolate (including by q-EVs) and analyze (including by TRPS) the EVs, e.g., https://pubmed.ncbi.nlm.nih.gov/20831623/;
https://www.ahajournals.org/doi/full/10.1161/CIRCRESAHA.117.309417;
https://www.cell.com/cell-reports-methods/pdfExtended/S2667-2375(21)00103-X). For instance, some of these papers do also provide "specific guidance on the peripheral blood draw component of human plasma isolation" and use q-EV SEC and TRPS. What is unique in the current manuscript that has not been highlighted in previous publications?

 We carefully assessed each article provided and have added the following paragraph to the introduction section. We addressed uniqueness and novelty specifically. 

“Myriad methods can analyze EVs and each may be appropriate for specialized applications.[40-42] For our purposes, however, a combination of automated SEC and TRPS works best. Because our protocol provides a simultaneous measurements of absolute quantification and size distribution of EVs 30-300 nm, applications like flow cytometry, NTA, and DLS, are not appropriate as detection is often limited to particles greater than 100 nm (or 60 nm for NTA/DLS) in diameter or do not provide absolute quantification.[43, 44] The uniqueness of this protocol lies in the way we leverage this combination of SEC and TRPS. Specifically, this protocol provides a method for reproducible quantification and is performed in a way that allows vesicles to remain intact throughout the measurement process, thus providing novel methods for normalization in any downstream application.”

Regarding the validation of the procedures, I would also be interested to know why other methods like NTA and DLS have not been used to validate the isolation and characterization methods obtained by TRPS.

 We appreciate that NTA and DLS are useful technologies and validation methodologies. However, our protocol is expressly built around TRPS for absolute quantification. Being that NTA and DLS do not quantify absolutely, we chose to maintain a singularly focused approach. 

Other comments:

Line 72: isn't ---- > is not

 Thank you for identifying this error, it has been corrected.

“Differential centrifugation protocols have been used many times, but this practice is not viable for several reasons having to do with the stress applied to vesicles by centrifugal forces (26).”

Line 86: is it 10um that is meant? Or rather 1um?

 The original citation contained a typographical error. This has been corrected to reflect the actual size range: 50nm – 1um.

“In both function and size, microvesicles are extremely heterogeneous: their size ranges between 50nm and 1µm and they shed into the extracellular milieu by the budding of the cell membrane (28).”

Lines 117-118: NTA is listed twice

 Thank you for identifying this error, it has been corrected.

“Several methods of EV quantification exist. These include resistive pulse sensing, nanoparticle tracking analysis, microscopy (scanning/transmission electron microscopy, and atomic force microscopy), dynamic light scattering (DLS), nanoparticle tracking analysis (NTA), flow cytometry, and small-angle X-ray scattering (SAXS) (27). 

Lines 125 and below: It is debatable to claim that TRPS is the most reliable and accurate method with sub-nm precision, and is superior to the more commonly used light scattering techniques. TRPS has, like other methods, limitations for heterogeneous EV populations, and the data obtained are directly impacted by the type of membrane selected to avoid clogging. Most people would think combining techniques is suitable for characterizing heterogeneous EV populations. Therefore I do not see the need to promote TRPS (or other Izon products) in this manuscript that describes protocols.

 To address concerns about perceived promotion of any one company’s product, we have amended the sentence such that it now states only the objective differences. 

“TRPS quantifies absolutely, whereas other commonly used light scattering techniques provide bulk estimates.” 

Lines 149-150: is there any risk of losing EV on the membrane of the centrifugal filter?

 This is an excellent point. The following statement has been added to clarify the purpose of centrifugal concentration of EVs.

“While we recognize some EVs may be lost during centrifugal concentration, this step was performed solely to further concentrate the samples so that even amounts of total protein (10 �g) are analyzed by western blot.”

Reviewer #2: Diehl and colleagues submitted a manuscript of protocol to explain the standardized method for the characterization of EVs in plasma.

This manuscript describes detailed procedures of plasma collection, EVs isolation, and sample procedures. In this paper, the explanation of the rationale for the selection of methods (SEC and TRPS). This paper is well-organized and has a logical basis.

If some requirements in the following comments are modified, this paper is going to be able to clear information for readers.

 Thank you for recognizing the importance of these protocols. We have addressed the following comments and enhanced the text’s overall clarity. We especially appreciate this reviewer’s careful attention to minute detail.

Minor comments.

1. Please add the information of reference at (DOI xx) line 148.

 We appreciate reviewer 2 pointing out our mistake, and we have rectified it by listing both follow protocol related DOIs:

(dx.doi.org/10.17504/protocols.io.3byl4jeb2lo5/v1, dx.doi.org/10.17504/protocols.io.ewov1ojnolr2/v1)

2. Spacing between number and unit (lines 84 and 86).

 Thank you for identifying the error, we have corrected the issue.

“EVs are divided into two groups: exosomes and microvesicles. Exosomes are commonly reported to be 30-120 nm in size and are released from multivesicular endosomes following fusion with the plasma membrane (28). In both function and size, microvesicles are extremely heterogeneous: their size ranges between 50 nm and 1 µm and they shed into the extracellular milieu by the budding of the cell membrane (28).”

3. Add unit at diameter (line 269 (Marfan 69.92))

 Thank you for identifying the error, we have corrected the issue.

“…Marfan 1.47 x 1011 particles/mL), mean diameter of EV particle (control 77.75 nm, Marfan 69.92 nm)…”

4. In terms of utilization of this technique, Marfan syndrome was selected to show the difference in the number and size distribution of EVs in plasma. Please add the physiological background information to explain why this syndrome could be a model as a diagnostic target.

 We have explained the physiological linkage between Marfan syndrome, aortic aneurysm, and extracellular vesicle concentration.

“We utilized the techniques described here to address a clinically meaningful question. In our system, we measure EV concentration and size in the plasma of healthy control patients and in Marfan patients who possessed a known thoracic aortic aneurysm. Previously, we have demonstrated that increased wall tension, as seen in aneurysm disease, leads to increased aortic fibroblast secretion of EVs. The primary cardiovascular complication associated with Marfan Syndrome is thoracic aortic aneurysm. Therefore, we are interested in determining whether EV concentration and size distribution can be leveraged as a diagnostic marker for aneurysm disease.”

---

## [Decision Letter · Decision Letter 1]

11 Apr 2023

A standardized method for plasma extracellular vesicle isolation and size distribution analysis

PONE-D-22-26362R1

Dear Dr. Natheniel,

We’re pleased to inform you that your manuscript has been judged scientifically suitable for publication and will be formally accepted for publication once it meets all outstanding technical requirements.

Kind regards,

Vibhuti Agrahari

Academic Editor

PLOS ONE

Additional Editor Comments (optional):

Reviewers' comments:

Reviewer's Responses to Questions

**Comments to the Author**

1. Does the manuscript report a protocol which is of utility to the research community and adds value to the published literature?

Reviewer #1: Yes

Reviewer #2: Yes

2. Has the protocol been described in sufficient detail?

To answer this question, please click the link to protocols.io in the Materials and Methods section of the manuscript (if a link has been provided) or consult the step-by-step protocol in the Supporting Information files.

The step-by-step protocol should contain sufficient detail for another researcher to be able to reproduce all experiments and analyses.

Reviewer #1: Yes

Reviewer #2: Yes

3. Does the protocol describe a validated method?

Reviewer #1: Yes

Reviewer #2: Yes

4. If the manuscript contains new data, have the authors made this data fully available?

Reviewer #1: Yes

Reviewer #2: Yes

**5. Is the article presented in an intelligible fashion and written in standard English?**

Reviewer #1: Yes

Reviewer #2: Yes

6. Review Comments to the Author

Reviewer #1: I thank the authors for responding to my questions and modifying the manuscript accordingly. I have no further comments

Reviewer #2: Thank you for revising the manuscript. Your protocol for exosome isolation and protein sample preparation is suitable to study the differences of EVs concentration between Marfan patients and control patients.

7. PLOS authors have the option to publish the peer review history of their article (what does this mean?). If published, this will include your full peer review and any attached files.

Reviewer #1: No

Reviewer #2: No
